# Inhibition of CXCR4 Enhances the Efficacy of Radiotherapy in Metastatic Prostate Cancer Models

**DOI:** 10.3390/cancers15041021

**Published:** 2023-02-06

**Authors:** Nisha Gupta, Hiroki Ochiai, Yoshinori Hoshino, Sebastian Klein, Jozef Zustin, Rakesh R. Ramjiawan, Shuji Kitahara, Nir Maimon, Despina Bazou, Sarah Chiang, Sen Li, Daniel H. Schanne, Rakesh. K. Jain, Lance L. Munn, Peigen Huang, Sergey V. Kozin, Dan G. Duda

**Affiliations:** 1Steele Laboratories for Tumor Biology, Department of Radiation Oncology, Massachusetts General Hospital and Harvard Medical School, Boston, MA 02114, USA; 2Institute of Pathology, University Medical Center Hamburg-Eppendorf, 20251 Hamburg, Germany; 3Department of Pathology, Massachusetts General Hospital and Harvard Medical School, Boston, MA 02114, USA

**Keywords:** prostate cancer, radiotherapy, SDF1α/CXCR4 signaling, primary lesions, bone metastases, orthotopic mouse models, vascular normalization

## Abstract

**Simple Summary:**

We examined the expression of SDF1α and its cognate receptor CXCR4 in human prostate cancer (PCa) lesions and the impact of CXCR4 inhibition alone and in combination with single-dose irradiation in orthotopic PCa models in mice. Both SDF1α and CXCR4 were highly expressed in primary and bone metastatic human PCa samples. Inhibiting CXCR4 activity in PCa cells abrogated SDF1α-induced invasion but did not sensitize them to irradiation in vitro. In orthotopic primary and bone metastatic PCa models, treatment with the CXCR4 antagonist AMD3100 alone was ineffective, but when added to radiotherapy, it significantly inhibited metastatic tumor growth. The mechanisms of AMD3100 included normalization of bone metastatic PCa vasculature. These results support testing SDF1α/CXCR4 inhibitors with radiotherapy in metastatic PCa patients.

**Abstract:**

Radiotherapy (RT) is a standard treatment for patients with advanced prostate cancer (PCa). Previous preclinical studies showed that SDF1α/CXCR4 axis could mediate PCa metastasis (most often to the bones) and cancer resistance to RT. We found high levels of expression for both SDF1α and its receptor CXCR4 in primary and metastatic PCa tissue samples. In vitro analyses using PCa cells revealed an important role of CXCR4 in cell invasion but not radiotolerance. Pharmacologic inhibition of CXCR4 using AMD3100 showed no efficacy in orthotopic primary and bone metastatic PCa models. However, when combined with RT, AMD3100 potentiated the effect of local single-dose RT (12 Gy) in both models. Moreover, CXCR4 inhibition also reduced lymph node metastasis from primary PCa. Notably, CXCR4 inhibition promoted the normalization of bone metastatic PCa vasculature and reduced tissue hypoxia. In conclusion, the SDF1α/CXCR4 axis is a potential therapeutic target in metastatic PCa patients treated with RT.

## 1. Introduction

Prostate cancer (PCa) is the most common malignancy in men. In most cases, the localized disease is typically indolent and curable by using surgical resection, androgen deprivation, and radiation therapy (RT) [1,2,3]. In contrast, metastatic disease is aggressive and becomes treatment-resistant over time. In patients with advanced castration-resistant PCa, bone metastasis is involved in most cases, and metastasis-directed RT is used for pain palliation and, in the case of oligometastases, for their long-term control [4,5,6]. Escalation of RT dose or biologically effective dose on all PCa lesions, made possible by recent technical advances in precision RT, is being developed to improve therapeutic outcomes.

In addition to optimizing RT, developing approaches targeting specific pathways in the PCa microenvironment to enhance tumor response to RT remains of great interest. One of these approaches is targeting the stromal-derived factor (SDF)-1α/CXCR4 pathway, which has been shown to enhance the progression of various tumors and their resistance to different treatments [7,8,9,10,11,12]. The SDF1α receptor CXCR4 inhibitor AMD3100 post-chemoradiation is currently being tested in a clinical trial in glioblastoma patients [13]. However, the interaction between the SDF1α pathway and RT in metastatic PCa remains incompletely characterized.

Previous studies have suggested a role for SDF1α and its receptor CXCR4 in PCa growth, metastasis formation, and response to RT. SDF1α is highly expressed in the bone marrow tissue and is crucial for the chemotactic recruitment of CXCR4-positive PCa cells and the formation of bone metastases [14,15]. CXCR4 is overexpressed in metastatic tumor tissues in PCa patients [16,17,18], and the metastatic growth correlates with higher levels of CXCR4 expression in primary tumor specimens [18]. A modest enhancement of the efficacy of fractionated RT by using AMD3100 has been demonstrated in several animal studies, including in ectopic (subcutaneous or intramuscular) PCa models [19,20]. Similar data were reported with AMD3100 and single-dose pre-irradiation in other cancer models [21,22,23]. In those models, the benefit of combination therapy was due, at least in part, to the inhibition of myeloid cell/macrophage infiltration and anti-vascular effects on tumor vessels post-RT [11,24]. However, the expression levels of SDF1α and CXCR4 in metastatic PCa post-RT, and their role in metastatic tumor regrowth after RT, remained unclear.

Here, we evaluated the role of the SDF1α pathway in PCa response to RT by studying human PCa tissues and preclinical models. We tested the impact of inhibiting CXCR4 using AMD3100 with single-dose RT in orthotopic (intra-prostate and intra-osseous) PCa models in mice to reproduce the primary and metastatic tumor microenvironments more faithfully.

## 2. Materials and Methods

### 2.1. Human PCa Tissue Samples

Clinically annotated tissue microarray (TMA) formalin-fixed, paraffin-embedded (FFPE) slides from patients with various stages of localized PCa were purchased from Nuclea Diagnostic Laboratories (Pittsfield, MA, USA). FFPE surgical samples from bone mPCa were obtained from the Institute of Pathology, University Medical Center Hamburg-Eppendorf (Hamburg, Germany); of these, 4 were from patients previously treated with RT for palliative purposes in a single dose of 8 Gy or 20 Gy given in 5 fractions about a month prior to resection. The slides were immunostained for CXCR4, CXCR7, and SDF1α, and the relative intensity of staining was analyzed with ImageJ software.

### 2.2. Cells

Human PCa cell line C4-2B was purchased from ATCC and cultured in DMEM/F12 (Lonza) supplemented with 10% FBS. Murine *Pten/Smad4-*null PCa cells were established from spontaneous tumors specifically developed in the prostate of *Pten/Smad4-*null mice [25], kindly provided by Dr. Ronald DePinho (MD Anderson Cancer Center, Houston, TX, USA). These cells were cultured in high-glucose DMEM (Cellgro) supplemented with 1% MEM non-essential amino acids (Invitrogen, Grand Island, NY, USA) and 10% FBS. For the engineering vessel experiments, human umbilical vein endothelial cells (HUVECs) were obtained from the Center for Vascular Excellence at the Brigham and Women’s Hospital (Boston, MA, USA) and cultured in endothelial growth medium (EGM, Lonza, Mapleton IL, USA) with manufacturer-supplied supplements. Murine 10T1/2 pericyte precursor cells (ATCC) were grown in DMEM supplemented with 10% FBS. Human pulmonary artery smooth muscle cells (SMCs) (Lonza) were cultured in smooth muscle cell growth medium-2 (SmGMTM-2) BulletKit (Lonza). Murine bone marrow-derived mesenchymal stem cells (MSCs) were harvested as described [26] and cultured in Iscove’s modified Dulbecco’s medium (IMDM) (Invitrogen) supplemented with L-glutamine plus 20% FBS.

### 2.3. Treatment Reagents and Irradiation

AMD3100 (Sigma-Aldrich, St. Louis, MO, USA) and recombinant (r)SDF1α were purchased from R&D Systems and used for in vitro and in vivo experiments. X-ray irradiation of PCa cells in vitro was performed using a vertical beam of XRAD 320 unit (Precision X-ray, North Branford, CT, USA) at 320 KV and filter 0.5 mm Cu, at a dose rate of 1.66 Gy/min.

### 2.4. Clonogenic Assay

To determine the radiosensitivity of cancer cell clonogens with or without CXCR4 treatment, we seeded appropriate concentrations of single-cell suspensions in T25 flasks with the intent to generate about 50–100 cell colonies/flask. Irradiation of cells (up to 6 flasks simultaneously) was performed 6 h later when cells were attached to the flask bottom; if used, AMD3100 was added to the flasks 1 h prior to RT or 10 min after RT. Colonies of 50 cells or more developed by surviving PCa clonogens (without changing the culture medium) were fixed, stained with crystal violet, and counted 7days later. To evaluate the treatment’s effects on cell growth, we measured the relative intensity of cell staining per flask using ImageJ software.

### 2.5. Migration Assay

We used the Boyden chamber assay to measure the trans-membrane migration of PCa cells. About 4 × 10^4^ cells were cultured in the upper chamber of each well of a trans-well plate (Corning) in the medium containing 10% FBS for 8 h. After the cells were attached to the bottom membrane (with an 8-um pore size), the medium was replaced with a serum-free one to starve the cells overnight. The next day, we added 100 ng/mL of rSDF1α with or without 5 µM of AMD3100 to the lower chamber of the trans-well plate, and the cells were incubated at 37 °C for 24 h. Then, the cells from the upper chamber were removed using cotton buds, and the cells which penetrated through the membrane in the low chamber were fixed, stained with crystal violet, photographed, and counted.

### 2.6. Western Blotting (WB) for Evaluating Protein Level Changes in PCa Cells, Mesenchymal Stem Cells, and Pericyte Precursor Cells after CXCR4 Inhibition In Vitro

In *PtenSmad4*-null cells, WB was used to analyze alterations in the expressions of several proteins associated with the epithelial-to-mesenchymal transition (EMT). Modulations in protein levels of pericyte marker NG2 and a key pro-angiogenic factor VEGF-A with AMD3100 and hypoxia were evaluated in MSCs and 10T1/2 pericyte precursor cells. Before WB analysis, the cells with or without AMD3100 were pre-incubated under regular/air (21% O_2_) and hypoxic (1% O_2_) conditions at 37 °C for 24 h; herewith, the hypoxia was generated using a hypoxic chamber.

### 2.7. Cell Co-Culture to Generate Vascular Network

HUVECs were co-cultured together with SMCs as optimized previously [27], with or without additional MSCs, at a ratio of 1:5:0.3 (respectively) in EGM-2 media (Lonza) in a 96-well plate. In this system, HUVEC-SMC co-culturing resulted in the formation of vessel networks without an exogenous matrix, and the addition of MSCs resulted in their differentiation into perivascular cells. The co-cultures were allowed 3 days to complete the vascular tube formation before adding AMD3100, and in 3 more days, the structures were stained for CD31 and NG2 and analyzed with immunofluorescence (IF) microscopy.

### 2.8. Orthotopic PCa Models

Orthotopic tumor xenografts were established by implanting 5 × 10^6^ C4-2B human PCa cells into the right dorsal prostate lobe of male nude mice aged 6–8 weeks. Orthotopic tumor grafts were established by implanting 2 × 10^5^
*Pten/Smad4-*null murine PCa cells into the right dorsal prostate lobe of male FVB mice aged 8–9 weeks. Tumor size was measured by caliper and/or intra-vital ultrasound imaging (Visual Sonics Vevo 2100 Micro-ultrasound, small animal imaging system, FUJIFILM VisualSonics Inc.), and the treatments were initiated when tumors reached about 4 mm in diameter, in both models. To establish orthotopic bone metastatic grafts, we implanted 2.5 × 10^5^
*Pten/Smad4-*null murine PCa cells 1:1 in 10 µL Matrigel (Corning Inc., Tewksbury, MA) in the tibiae of FVB or F1 (FVB x C57Bl/6) male mice of 8–10 weeks of age. Bony tumors were measured with a 1–2-day interval using a caliper, and the treatments were commenced when the leg thickness, including the tumor, reached 8 mm on average in two orthogonal dimensions across the bone (the thickness of tumor-naive legs was about 4 mm). All animal experiments were conducted according to a protocol approved by the Institutional Animal Care and Use Committee of the Massachusetts General Hospital, Boston, MA, USA.

### 2.9. In Vivo Irradiation and AMD3100 Treatment

Local tumor irradiation was implemented using a double-collimated vertical beam of XRAD 320 unit (320 kV, 12.5 mA, filter 0.5 mm Cu) with a dose rate of 3.52–3.75 Gy/min. Mice bearing tumors in the prostate were anesthetized with ketamine/xylazine, i.p., gently secured with tape in the supine position, and placed under a secondary collimator, lead shield with a hole of 15 mm in diameter located above the tumor. Tumors implanted in the tibia were irradiated in a 30-mm collimated field, herewith stretching the leg of conscious mice secured in a jig. A dose of localized irradiation delivered to the tumors was 12 Gy unless otherwise specified. For CXCR4 inhibition, mice were treated with 10 mg/kg AMD3100 (Sigma-Aldrich) dissolved in water and using Alzet osmotic pumps (Durect, CA, USA) for its continuous delivery at a rate of 0.25 µL/h for 14 days [28]. The pumps were implanted dorsolaterally under the skin. When the treatments were combined, tumor irradiation was immediately followed by the pump implantation. The pre-specified endpoints were the size of the tumors at certain time points (prostate tumors) or the time taken for the tumors to grow to a cut-off size of 13 mm or mouse death (bony tumors). The developments of lymph node and lung metastases were also assessed in the same mice.

### 2.10. Immunofluorescence (IF)

At the experimental endpoint, tumors were surgically resected and immersed in 4% formaldehyde in PBS (time of fixation in hours = tumor radius in mm). This was followed by incubation in 30% sucrose in PBS overnight at 4 °C and subsequent mounting in freezing media (OCT, Tissue-Tek, Torrance, CA, USA). Twenty-μm thick frozen sections were air-dried at room temperature for 1 hr, then post-fixed in acetone at −20 °C for 5 min, and then washed in PBS. For blocking, 5% normal horse serum (Jackson Immunoresearch) was used (1 h at room temperature), and for diluting primary antibody (for the endothelial marker CD31, pericyte marker desmin, or hypoxia marker CA-IX, applied overnight at 4 °C) and secondary antibody (1 h at room temperature in darkness). Sections were mounted with 4’-6-diamidino-2-phenylindole–containing mounting media (Vectashield, VectorLabs, Burlingame, CA, USA). The list of primary and secondary antibodies used in the study for staining various proteins in different samples and assays is presented in Appendix A. Co-culture and tumor tissue imaging was performed using an Olympus FV1000 confocal microscope. The stained areas were quantified for vessel density and pericyte-endothelial cell proximity using an in-house MATLAB algorithm [29]. The fraction of vessels with open lumens was analyzed as previously described [30].

### 2.11. Statistical Analysis

Comparison between two experimental groups was performed using an unpaired two-tailed Student’s t-test. One-way ANOVA followed by Tukey’s post hoc corrections for multiple comparisons was used when comparing more groups. Two-way ANOVA was additionally used to determine the significance of the overall effect of either RT or AMD3100 in four-group treatment experiments. Median times to the pre-specified endpoint (certain tumor size or mouse death) were estimated by Kaplan–Meier curves and compared using the long-rank (Mantel–Cox) test. Fisher’s exact test was used to compare the proportions of mice with metastases. Differences were considered statistically significant if *p*-values were less than 0.05. In all figures, * *p* < 0.05, ** *p* < 0.01, *** *p* < 0.001, **** *p* < 0.0001. The error bars of mean values depict SEMs in all preclinical studies and SDs for the clinical ones.

## 3. Results

### 3.1. SDF1α Expression Is Increased with Tumor Progression and after RT in Human PCa Tissues

We first evaluated the expression of SDF1α and its receptors CXCR4 and CXCR7 in a commercially available TMA that included samples of resected primary PCa. Semi-quantitative IHC analyses in these formalin-fixed, paraffin-embedded PCa tissues showed that both SDF1α and CXCR4 were expressed in these tumors (Figure 1a). The most homogeneous and intense staining was seen for CXCR4, regardless of Gleason scores. However, the expression of SDF1α was more intense in PCa tissues from advanced (Gleason scores 7–10) versus less advanced (Gleason scores 3–6) diseases. We also detected a differential distribution for the expression of these markers: CXCR4 was primarily and uniformly expressed in PCa cells, while SDF1α expression was seen predominantly in the tumor stroma. CXCR7 was also largely expressed in the tumor stroma but was not associated with the Gleason scores.

We next also analyzed the expression of SDF1α and CXCR4 in bone metastatic PCa tissues, some of which were irradiated less than a month before surgery for palliation (Figure 1b and Appendix A). We found that SDF1α was expressed in both non-irradiated parenchyma and stroma, and the staining intensity in the stromal component was remarkably increased post-RT. At the same time, CXCR4 expression was largely restricted to cancer cells in these lesions and appeared unchanged after irradiation.

Collectively, these correlative findings indicate that CXCR4 is highly expressed in both primary and metastatic PCa and that higher SDF1α levels in the stroma may be associated with previous irradiation in metastatic lesions. These observations are consistent with the clinical literature and motivated us to further examine the roles of SDF1α/CXCR4 inhibition with RT in vitro and then in vivo using orthotopic primary and bone metastatic PCa models in mice.

### 3.2. Inhibition of CXCR4 Decreases Invasion of PCa Cells and Has Only a Growth Inhibitory but Not Radiosensitization Effect In Vitro

In the three PCa cell lines studied, we showed that inhibition of CXCR4 with AMD3100 significantly reduced their migration in Boyden chamber assay in response to (r)SDF1α (*p* < 0.01) (Figure 2a). These data indicated that SDF1α/CXCR4 signaling might be important for stimulating PCa cell motility. As enhanced migration is usually associated with epithelial-to-mesenchymal transition (EMT) phenotype, we also assessed the effect of CXCR4 inhibition on EMT-related genes. Using *PtenSmad4*-null cells, we found deregulation of EMT markers—an increased expression of E-cadherin and inhibition of vimentin, Snail, Slug, and ZEB1—following pharmacological inhibition of CXCR4 with AMD3100 (Figure 2b and Appendix A).

On the other hand, when combined with RT, CXCR4 inhibition had no effect on PCa cell radiosensitivity (Figure 2c–f). AMD3100, either alone or in combination with 8 Gy, slightly decreased, in a dose-dependent manner, the number of *PtenSmad4*-null PCa cell colonies. However, this minor (≤20%) decrease in colony count was likely due to the inhibition of cell growth by AMD3100. The combined effect was seen irrespective of whether CXCR4 inhibition was started prior to or after RT.

### 3.3. CXCR4 Inhibition after RT Delays the Regrowth and Metastasis of Orthotopic PCa Xenografts

To study the efficacy of CXCR4 inhibition and local single-dose RT on primary PCa tumor growth, we used an orthotopic PCa xenograft model–human C4-2B in immune-deficient nude mice. All treatments were initiated when tumors reached a volume of approximately 20 mm^3^. First, we tested these tumors’ dose-dependent response to RT and metastases’ development. The autopsy of all mice on day 28 showed a clear decrease with RT dose of both primary tumor size and frequency of metastasis in the non-irradiated lymph nodes (Appendix A). We detected no overt toxicity to normal tissues following RT.

Next, we treated mice bearing established primary PCas with local RT at a dose of 12 Gy, AMD3100 delivered continuously via osmotic pumps for two weeks, or their combination versus control (PBS osmotic pumps). When evaluated at day 28, CXCR4 inhibition with AMD3100 treatment alone had no effect on the primary tumor growth and showed just a tendency to enhance the growth inhibition induced by RT (Figure 3a). RT significantly inhibited both primary tumor growth and lymph node metastasis (Figure 3b).

In separate experiments, mice were treated with local 12 Gy RT of the tumor, with or without systemic CXCR4 inhibition, and followed for a longer period of observation. All mice were sacrificed at day 60 to evaluate the longer-term post-radiation effects on primary tumor regrowth and metastatic progression. We found that combining RT with the two-week AMD3100 treatment induced a modest but significant inhibition of primary tumor growth compared to RT alone at this late time point (Figure 3c). Moreover, this combination therapy significantly decreased the incidence of lymph node metastases from irradiated primary PCas at day 60 (Figure 3d).

### 3.4. CXCR4 Inhibition after RT Delays the Regrowth of Established Bone Metastatic PCa

We then examined the efficacy of combining CXCR4 inhibition with RT in the syngeneic mouse model of bone metastatic PCa, using *Pten/Smad4*-null cells implanted in the tibia of F1 mixed background (FVB × C57Bl/6) mice. Treatments were initiated when tumors were well established, i.e., at approximately 8 mm of the combined thickness of the leg plus the tumor. In this syngeneic model, which was more aggressively growing and radioresistant than C4-2B xenografts, we found that treatment with AMD3100 alone induced only a non-significant tendency for delayed growth of bone metastatic PCa lesions. However, when combined with RT (12Gy, which was modestly effective alone), AMD3100 produced statistically significant increases in inhibition of tumor growth and survival (*p* < 0.05) (Figure 4a,b). As determined at autopsy at the experimental endpoint (when a tumor reached the pre-specified size or when a mouse became moribund), most mice had spontaneous, “secondary” metastases in the lung (Appendix A) and occasionally liver metastases (not shown), with no significant difference between groups.

### 3.5. CXCR4 Inhibition Normalizes Vessel Structure and Reduces Hypoxia in Bone mPCa

We next examined the changes induced by CXCR4 inhibition on the tumor microenvironment of PCa. To address the question of AMD3100-induced alterations on the tumor stroma, and more specifically on tumor vasculature, we implanted *Pten/Smad4*-null cells into either prostate or tibia in FVB mice. When established, the tumors were treated with a single-dose local RT (8 Gy) alone, AMD3100 (osmotic pumps) alone, or RT followed by AMD3100. All mice were sacrificed, and tumor tissues were collected after four days of treatment. Using IF for CD31 (endothelial cell marker), desmin (perivascular cell markers), and CA-IX (hypoxia marker), we found a site-dependent effect on tumor vasculature and hypoxia in response to CXCR4 inhibition, with statistically significant changes or strong trends only in the metastatic PCa lesions (Figure 5). In this model, AMD3100 treatment did not affect vessel density (Figure 5).

However, double IF analysis for CD31 and desmin revealed that CXCR4 inhibition increased the fraction of mature vessels (covered with pericytes) (*p* = 0.03, two-way ANOVA). In addition, AMD3100 treatment significantly increased the fraction of vessels with an open lumen and reduced tissue hypoxia (measured by IF for CA-IX). These changes in the bony metastatic PCa model were associated with a decreased SDF1α expression in the tumor tissue following AMD3100 treatment (*p* = 0.002, two-way ANOVA) (Appendix A). When implanted in the prostate, the resulting PCas showed only a tendency for a change in these structural and functional parameters after AMD3100 treatment, which did not reach statistical significance (Appendix A). Of note, the addition of RT did not significantly affect all these parameters at this time point in either model. Together, our analyses of vessel structure and markers of hypoxia show that CXCR4 inhibition differentially promoted the normalization of tumor vasculature in bony metastatic versus primary *PtenSmad4*-null murine PCa lesions.

### 3.6. CXCR4 Inhibition Can Induce Mesenchymal Stem Cells (MSC) and Pericyte Precursor Cell Differentiation into Perivascular Cells

To examine the potential mechanisms by which CXCR4 inhibition promoted vascular normalization/maturation in mPCa, we performed in vitro experiments focused on perivascular cells. We first evaluated the direct effect of CXCR4 inhibition with AMD3100 on 10T1/2 pericyte precursor cells and bone marrow MSCs under normoxic and hypoxic conditions. Western blot analysis showed elevated protein levels of pericyte marker NG2 in both MSCs and 10T1/2 cells after CXCR4 inhibition, but only when the cells were cultured in hypoxic conditions (Figure 6a and Appendix A).

Using 10T1/2 murine pericyte precursor cells, we also evaluated AMD3100-induced changes in the expression of VEGF-A, a key pro-angiogenic factor, and found that VEGF-A protein level also increased after AMD3100 treatment under hypoxic culture conditions only (Appendix A), but not in normoxic conditions. These observations suggest a key role for hypoxia, which is often seen in bone metastatic PCa, on the effect of CXCR4 inhibition on mesenchymal precursors.

To further examine the effects of CXCR4 inhibition on vessels, we used an in vitro endothelial tube formation assay. We found that CXCR4 inhibition with AMD3100 in a co-culture system using HUVECs and SMCs slightly promoted the formation of vessel-like structures, increasing the area occupied by CD31-positive endothelial cells by ~20%, as compared to control (untreated) co-culture.

As a significant effect on vessel maturation after treatment with AMD3100 was only seen in the bone metastatic PCa model, we then investigated whether CXCR4 inhibition can also affect the differentiation of bone marrow MSCs into perivascular cells. To this end, we added MSCs to the standard HUVEC-SMC co-culture to mimic vessel formation in the mPCa environment. After three days of incubation with AMD3100, we detected an increased co-localization of MSCs with endothelial cells (Figure 6b–d), indicative of functional MSC differentiation into pericytes in the vessel-like structures.

Collectively, these results show that CXCR4 inhibition can promote vessel stabilization in hypoxic conditions.

## 4. Discussion

RT is being increasingly delivered at higher doses per fraction for PCa treatment for both primary and bony metastases. Despite the favorable local control rates achieved in less advanced PCa, a reinforcement of high dose RT is clearly needed in high-risk cases [3]. Our preclinical studies were motivated by the high expression levels of SDF1α and its receptor CXCR4 in human PCa and mPCa specimens and the increased SDF1α expression in more advanced primary PCa and mPCa stroma after irradiation. These clinical findings are consistent with the results from preclinical studies using irradiation in orthotopic brain tumor models [22]. In addition, preclinical studies have also demonstrated the efficacy of SDF1α axis-related modulation of RT in primary PCa and other tumor models in rodents [19,20,21,22,23,31]. These studies used the clinically available CXCR4 inhibitor AMD3100 combined with RT, but the PCa was modeled in ectopic (subcutaneous) grafts, which do not reflect the stroma of primary or metastatic PCa lesions [19,20]. Here, we tested the hypothesis that inhibition of the SDF1α/CXCR4 axis can increase the efficacy of SBRT-like treatment in orthotopic models of primary and metastatic PCa.

As seen in the clinical specimens, SDF1α and its receptors were expressed in the PCa cell lines used in our in vitro and in vivo experiments. In these PCa cell lines, pharmacologic inhibition of the CXCR4 receptor reduced the migration of PCa cells in response to exogenous SDF1α. In addition to changes in cell phenotype, we also detected a modest growth inhibition by AMD3100 in *PtenSmad4*-null murine PCa cells, with or without RT. However, in these experiments, the CXCR4 inhibition did not change the killing of PCa clonogens by RT. This finding agrees with a prior report that showed that signaling through CXCR4 did not modulate radiation-induced DNA damage and repair in PCa cells [20]. Collectively, these data suggest that AMD3100 affects long-term PCa cell viability through mechanisms unrelated to radiosensitization.

Indeed, a prior study demonstrated the critical role of CXCR4 inhibition post-RT: A three-week course of AMD3100 treatment was more effective in inhibiting tumor growth when used after versus concurrently with a three-week regimen of fractionated RT with cisplatin [32]. Thus, we tested the impact of AMD3100 treatment initiated after single-dose RT of the primary PCa, as previously described [21,22,23]. Our study results show that, although AMD3100 alone is ineffective, it potentiates the antitumor effect of single-dose 12 Gy RT in primary PCa models. This local tumor control benefit was modest. However, when we performed a longer-term experiment in primary PCa xenografts treated with RT, AMD3100 potentiated the local control and significantly decreased the spontaneous lymph-node metastasis. Moreover, in an orthotopic (intraosseous) murine mPCa model, combining RT and AMD3100 treatment significantly delayed irradiated tumor growth and prolonged mouse survival.

Previously, we and others have reported that the antitumor efficacy of CXCR4 inhibitors combined with RT is mediated by effects on myeloid cells [11,21,22]. Here, we report an unexpected effect of CXCR4 inhibition in the context of the rapid normalization of PCa vessels in bony metastases. While the density of mPCa vessels in this model was unchanged after four days of AMD3100 treatment (with or without RT), the vessels were more mature (i.e., showed higher coverage by pericytes) and had open lumens (as opposed to collapsed), which associated with decreased tumor hypoxia. Anti-CXCR4 treatment-induced vascular normalization and reduced hypoxia could reprogram the tumor immune microenvironment [33] and synergize with the direct effects of AMD3100 on immunosuppressive myeloid and regulatory T cells [34,35]. Of note, the same combination treatment (AMD3100/RT) did not significantly affect the tumor vessels or tissue hypoxia in the intra-prostate lesions using the same *PtenSmad4*-null PCa model. These divergent effects of AMD3100 in the same model but in different organ sites/microenvironments may explain in part the inconsistent results on the role of CXCR4 inhibition in metastasis from different animal models [20,31,34,36].

Vessel normalizing effects following AMD3100 alone were previously reported in other models [19,24], while other studies showed anti-vascular/vessel pruning effects after CXCR4 inhibition [15,22,35,37]. Silencing of CXCR4 was shown to reduce VEGF expression in PCa cells and xenografted tumors [37,38], suggesting an indirect mechanism of AMD3100-induced vessel normalization in mPCa. Since SDF1α can function as a pro-angiogenic molecule via several mechanisms [12,39], the decrease in SDF1α expression after AMD3100 treatment in mPCa could also be involved in the tumor vessel normalization detected in this model. To better understand the mechanism by which AMD3100 treatment-induced increased pericyte coverage of bone mPCa, we tested the impact of CXCR4 inhibition on perivascular precursor cells in vitro. We found that CXCR4 inhibition induced a pericyte-like phenotype in bone marrow-derived MSCs cultured in hypoxic conditions to mimic the mPCa microenvironment. Furthermore, in a tube formation assay, we also found that the co-cultured MSCs co-localized with endothelial cells suggesting a functional differentiation of bone marrow-derived MSCs into perivascular cells. These data are consistent with studies demonstrating that bone marrow-derived MSCs could function as pericyte precursors and contribute to new vessel formation [19,40]. In contrast to the reduced VEGF-A expression effect seen after CXCR4 inhibition in PCa cells cultured in normoxia [38], we found that AMD3100 treatment induced a slight increase in VEGF expression in 10T1/2 mesenchymal precursor cells cultured in hypoxic conditions. Future studies should reveal the impact of these differential effects of CXCR4 inhibition on stroma versus cancer cells, in different microenvironmental conditions, on tumor progression, vascular structure, and function. Future studies should also define the time window of vascular normalization induced by CXCR4 inhibition in bony mPCa, to allow the scheduling of SBRT during the periods of decreased tissue hypoxia. The efficacy of this strategy has been demonstrated preclinically using angiogenesis inhibitors [41,42].

## 5. Conclusions

In summary, SDF1α and its receptor CXCR4 are highly expressed in human PCa tissues and associated with disease progression. Inhibiting CXCR4 reduced irradiated PCa cell migration and had modest growth inhibitory effects in vitro. However, in vivo treatment with the CXCR4 inhibitor AMD3100 combined with single-dose RT significantly inhibited metastatic growth in orthotopic PCa models. The benefit in bony mPCa was associated with vessel normalization after AMD3100 treatment. These novel insights into the effects of CXCR4 inhibition in the context of RT should be considered to optimally combine it with RT for the therapy of patients with advanced PCa.

## Figures and Tables

**Figure 1 cancers-15-01021-f001:**
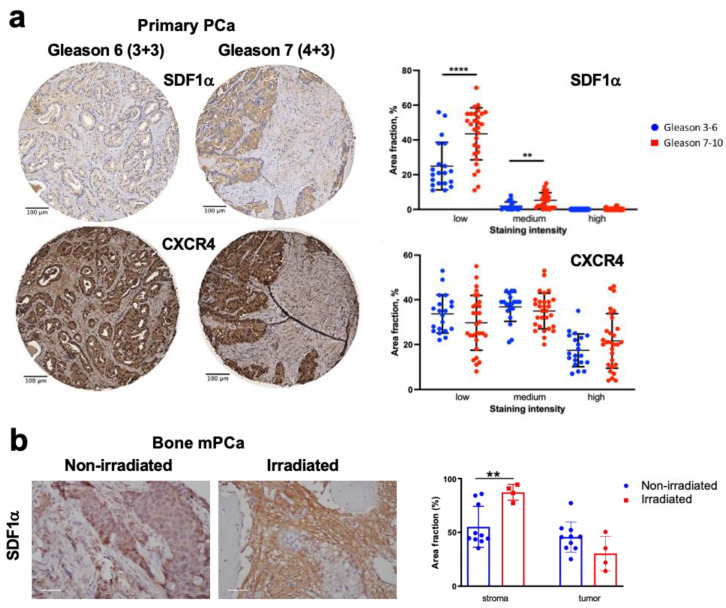
Expression of SDF1α and CXCR4 in primary and bone metastatic prostate cancer (mPCa) tissues. (**a**) Representative images and quantitative analysis of SDF1α and CXCR4 immunostaining in primary PCa tissue, stratified by Gleason score (Gleason 3–6: *n* = 20, median score 6, median age 76 ± 9 years vs. Gleason 7–10: *n* = 29, median score 8, median age 78 ± 6 years) and staining intensity (high/medium/low) analyzed with ImageJ software using predefined thresholds for the respective categories. (**b**) Representative images and quantification of SDF1α expression by immunostaining in metastatic PCa tissue, radiation-naïve, or previously irradiated less than a month prior to surgery; scale bar: 50 µm. ** *p* < 0.01; **** *p* < 0.0001.

**Figure 2 cancers-15-01021-f002:**
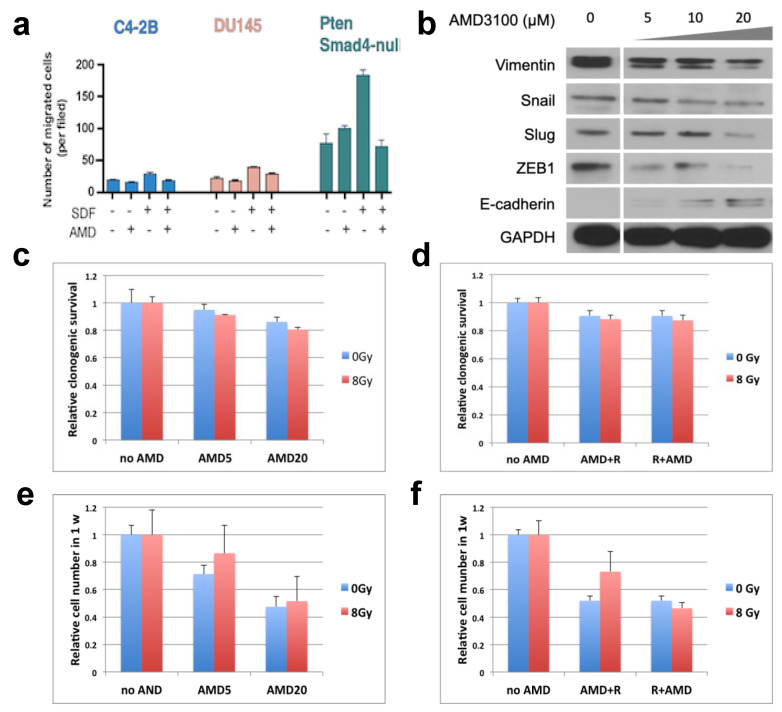
Effects of CXCR4 inhibition with AMD3100 (AMD) on PCa cell migration, expression of EMT-related genes, and radiosensitivity in vitro. (**a**) AMD3100 inhibited PCa cells’ migration, with or without stimulation with rSDF1α. (**b**) Changes in markers associated with EMT in *PtenSmad4*-null murine PCa cells following CXCR4 inhibition with AMD using concentrations between 5–20 µM. (**c**–**f**) Effects of AMD3100 (5 or 20 µM) on clonogenic survival (**c**,**d**) and the total number of viable cells (**e**,**f**) treated with and without irradiation at a dose of 8 Gy (which decreased clonogenic survival by 13% compared to control); colonies were stained and analyzed 1 week after treatment initiation. AMD was administered at concentrations of 5 and 20 µM 1 hr before irradiation (**c**,**e**) and at a concentration of 20 µM 10 min after irradiation (**d**,**f**).

**Figure 3 cancers-15-01021-f003:**
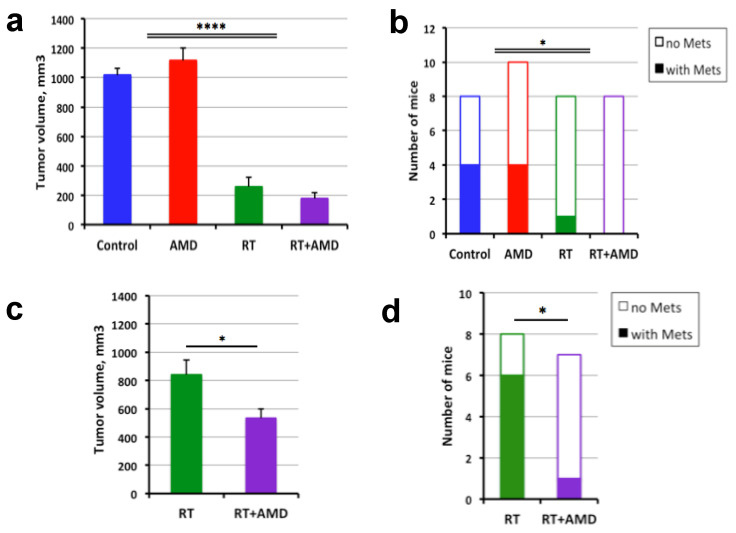
Efficacy of CXCR4 inhibition added to RT in nude mice bearing orthotopic human C4-2B PCa xenografts. (**a**,**b**) Tumor volume (**a**) and the fraction of mice with LN metastasis (**b**) 28 days after radiotherapy (RT) at a single dose of 12 Gy and/or the initiation of 2-week AMD3100 (AMD) treatment. (**c**,**d**) The long-term effects of combination therapy on primary tumor volume (**c**) and LN metastasis (**d**), in the same model, at day 60. * *p* < 0.05; **** *p* < 0.0001.

**Figure 4 cancers-15-01021-f004:**
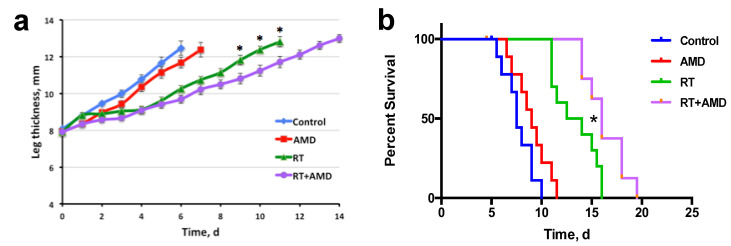
Therapeutic efficacy of post-radiation inhibition of CXCR4 signaling in C57Bl/6 x FVB F1 chimeric mice bearing syngeneic *PtenSmad4*-null PCa implanted in the tibia (mPCa model). (**a**) The dynamics of leg thickness (including tumor) following radiotherapy (RT) at a single dose of 12 Gy and/or the initiation of a 2-week AMD3100 (AMD) treatment, compared to control. (**b**) Kaplan–Meier survival distributions in the four treatment groups. * *p* < 0.05.

**Figure 5 cancers-15-01021-f005:**
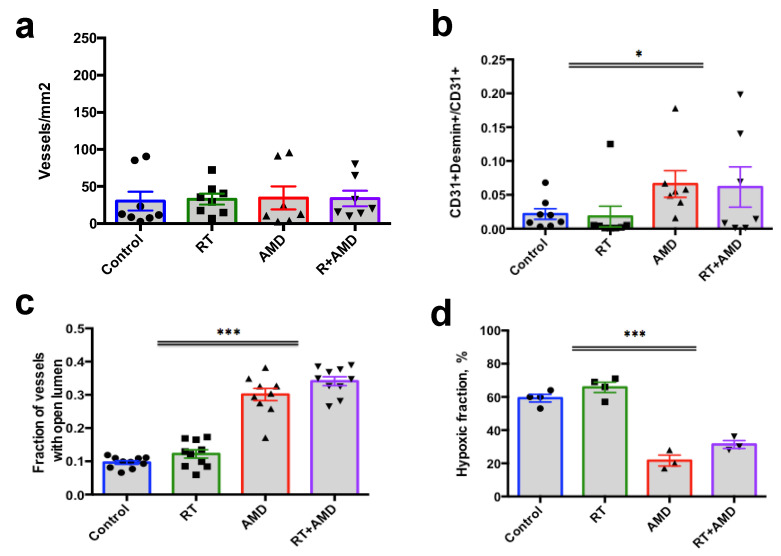
Effects of AMD3100 on tumor vasculature in bony PtenSmad4-null mPCa tissue evaluated by immunofluorescence (IF). (**a**–**c**) The density of vessels (**a**), the fraction of mature vessels (**b**), and the fraction of vessels with open lumens (**c**) were measured by IF for CD31 and desmin. (**d**) Expression of the hypoxia marker CA-IX measured by IF. Tissue collection for analysis was performed 4 days after radiotherapy (RT) and/or AD3100 treatment. * *p* < 0.05; *** *p* < 0.001.

**Figure 6 cancers-15-01021-f006:**
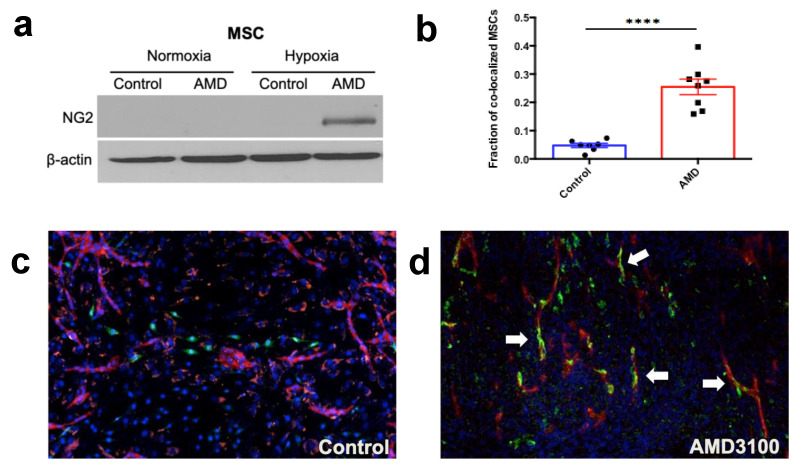
Effects of CXCR4 inhibition with AMD3100 on phenotype of MSCs and their vessel stabilizing action in vitro. (**a**) Increased expression of pericyte marker NG2 following incubation of MSCs under hypoxic (1% O_2_) but not air conditions. (**b**–**d**) Increased co-localization of MSCs with endothelial cells in HUVEC-SMC co-culture after incubation with AMD3100 (**b**). Representative fluorescence microscopy of co-cultures in the control (**c**) and AMD-treated (**d**) groups. Arrows show the perivascular MSCs. In blue, DAPI counterstaining; in red, CD31; and in green, NG2. **** *p* < 0.0001.

## Data Availability

Not applicable. All data are included in the manuscript.

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
