# Peer review of "Inhibition of CXCR4 Enhances the Efficacy of Radiotherapy in Metastatic Prostate Cancer Models"

_cancers, 2023, doi:10.3390/cancers15041021_

Round 1
Reviewer 1 Report
The authors present a preclinical study on SDF-1/CXCR4/CXCR7 signalling and radiation response in orthotopic prostate cancer and bony metastasis. Overall, the data appears to support as shown in other model systems that this pathway might represent a rational target to overcome treatment resistance.
Some minor remarks for possible consideration:
1. Abstract: “Single-dose radiotherapy (RT) is a standard treatment for advanced prostate cancer (PCa) 30 patients.” This is misleading and wrong. Furthermore it doesn’t fit the statements of motivation in the introduction. The authors should refrain on speculating about dose per fraction effects and clinical implications. They have used single dose in the experiments. Fine. They did not use fractionated Rt as comparator. Therefore the respective parts in intro and discussion should focus on the topic and not speculate on time dose fractionation in clinical radiotherapy. It appears more straight forward to discuss targeting approaches and translational challenges related to this.
2. Fig1A: Staining intensity was used for splitting the data in three groups and make GS comparisons within these groups. What is the rationale for doing that? It appears important to show also the association of SDF expression independent from arbitrary intensity classification. GS is mostly used to separate GS 8-10 from the rest as these scores indicate biologically aggressive behavior. GS 7 is heterogeneous and would require further information (compare ISUP classification). If the clinical data is not strong enough to support the rationale than it should be taken out as the pathway as such has been described as relevant and it is not essential to justify the experimental study. It appears unlikely that the authors after the results from figure 1 decided to start the project. Rephrasing should be considered.
3. Fig2A: data points in SF over dose figures should not be connected but fitted and the function graphically displayed
4. Fig3 A and B: not entirely clear what is actually shown here. Text and legend appear not to match. What explains the difference between A and B?
5. Fig4a: If growth delay is the endpoint then not volumes at a given time should be compared but e.g. the time to grow to 12mm.
6. Fig4b: Which groups were compared by the log rank test? RT versus RT+AMD? The effect of 2 days difference in mean survival appears very small for being statistically significant.
Author Response
Please find the responses attached.

Reviewer 2 Report
Report on the manuscript untitled “Inhibition of CXCR4 enhances the efficacy of radiotherapy in metastatic prostate cancer models” by Gupta N. et al.
In this work, the authors evaluate the combination of the downregulation (shRNA) or inhibition (AMD3100) of CXCR4 and radiotherapy on different prostate cancer models and conclude that this combination has a modest radiosensitizing effects in vitro, inhibits metastatic growth in orthotopic primary and metastatic prostate cancer models and reduced lymph nodes metastasis. Moreover, CXCR4 inhibition normalizes vasculature structure and function.
These conclusions are based on well-illustrated experiments performed on several prostate cancer (PCa) models: from biopsies of patients, in vitro human and mouse cell lines, followed by in vivo experiments on orthotopic models or on bone metastatic PCa. Then, the authors explored the vascular normalization and maturation in a co-cultured cell system.
The impact of the inhibition of CXCR4 combined with radiotherapy has been previously described but never on orthotopic and metastatic PCa models. So, this work is of interest and is very important for a potential targeting / biological approach to PCa patient treatment in the field of personalized medicine.
However, as the main subject of this article is based on radiation response following treatment, the in vitro radiotherapy section should be done more carefully.
There is also a lot of small mistakes, lack of precision or consistency in the figure legends that make sometimes the reading of this article difficult to follow.
Major revisions:
- Regarding the in vitro radiosensitizing effects of the inhibition of CXCR4, the combination treatment has been described in the Materials & Methods (M&M) section based on clonogenic or on MTT assays.
There is no explanation which curves (figures 2D and Figure 2E) resulted from MTT or clonogenic assays (except may be for the AMD3100 inhibition ± radiation described the main text). Does-it mean that the figure 2D results from MTT assay? The figure 2D-E legend is not correct and not precise enough. We don’t know which assay has been chosen, how many experiments were done (n=?) and if statistical analysis has been performed. Please can you provide all these information either in the M&M section or in the Figure 2 legend?
Line 278 : “we used two standard assays (MTT) and clonogenic assays”.
Only clonogenic assay is considered as the gold standard method to assess the radiation response. MTT assay (widely used to test drug response on cells) is not a standard method in radiobiology and could not be used to determine a potential radiosensitizing effect as radiation is a mitochondrial biogenesis inducer (Rai Y. et al, “Mitochondrial biogenesis and metabolic hyperactivation limits the application of MTT assay in the estimation of radiation induced growth inhibition”, Scientific Reports, (2018) 8: 1531. DOI: 10.1038/s41598-018-19930-w). As the radiation-response curves are based on relative survival of irradiated versus non-irradiated samples, the use of this method induces a bias and leads most of time to a wrong conclusion.
Can the authors provide radiation-response curves resulting only from clonogenic assays on all the cell lines? If clonogenic assay is not possible on all the cells considered in this study (some cells do not form colonies), the proper experiments should be to count the cells after the same doubling time of each condition following irradiation (at least 5 population-doubling time) to be able to compare all the PCa cells passing through the same number of mitosis (this could be different from one cell line to another). Can the authors provide clonogenic or growth inhibition on these PCa models?
- All the experimental data from the radiation-response curves should be fitted with the classical linear-quadratic equation: S=exp(-aD-bD2), where S is the survival fraction and D the dose (in Gray). From this fit, the alpha and beta parameters can be extracted allowing the calculation of the D10 (dose to achieve 10% survival) for example. Then statistical analysis can be done comparing these D10 doses from non-treated to treated samples on at least three different experiments. Using this method may be the radiosensitizing effect of the inhibition of CXCR4 could be highlighted (or not) with a proper conclusion and robust statistics. Can the authors provide such fitted curves and D10 calculation and comparison?
- The dose delivery using an XRAD320 system is very challenging, especially when using an orthotopic prostate irradiation in mice.
Can the authors provide (in the main text or in the supplementary section) more information on the dosimetry and what could be the impact of the 15 mm collimation to the overall mice survival due to the irradiation of the bladder or the gut?
How many mice did you use per arm? How many experiments did the authors perform? Can the authors provide these data?
- lines 295-305 : The results of the 3.3 section and the Figure 3 legend are not consistent and not clear at all.
Line 298 “… showed no effect on the primary tumor growth…” : are these data not shown?
Line 299 “ … showed just a tendancy to enhance the growth inhibition induced by RT (Figure 3A)…” : How the authors can conclude with a “tendancy” and show a statistical analysis with one star between the two groups in Figure 3A?
Line 301: The Figure 3B refers in the main text as the results on lymph node metastasis. This is not in accordance with the Figure 3B legend. Can the authors clarify this point? Again, how many mice did the authors used in each arm (primary or metastatic)?
Idem for the Figure 3C described as a tumor growth inhibition in the main text (line 310) and as a “number of mice” in the Figure. This not clear which Figure section refers to primary or metastatic PCa models.
Can you please re-organize the Figure 3, give the proper information at the right place, and bring more precision in the Figure legend?
- Lines 459-460: Can the authors precise in which cell line(s) Wang et al. performed their experiments to comment the differences observed with your results regarding the VEGF-A expression?
- Lines 402-404 : The authors described the work of Domanska U. et al, referring only to their ectopic PCa graft. Even if the orthotopic PCa model is more relevant to study the in vivo radiation response, these authors found also an in vitro radiosensitizing effects following AMD3100 treatment on two different PCa cells using clonogenic assay, with a modulated response based on the presence of the stroma or not. Can the authors comment this work in the discussion in the light of your results?
- Line 106 and line 204: Table S1 and the primary and secondary antibody list is not provided. Can the authors show and add these two supplementary data?
Minor revisions:
- As the main parameter of all this work is based on CXCR4, can the authors put in Figure 1 the CXCR4 images and quantification provided in FigureS1 instead of in the supplementary section?
- Figure numbers in the main text are sometimes in bold. Can the authors be consistent with or without bold labels in all the text?
- Figures captions are written in capital letters in the text and not in the legends or in the figures (Figures 3-6). Can the authors standardize the labels?
- Line 102 ; a “) “ is missing
- Line 178 : “of” instead of “off”
-
Figure 1 : “primary PC” : Do you mean PCa?
BMPC : can you explain in the legend what BMPC means?
- Lines 234-240 and in supplementary section. Very often the label “alpha” is missing following SDF1
- Figure 1A : which criteria did you choose to classify between “low/medium and high” the SDF1a immunostaining?
- line 278 : “clonogenic” instead of “clonogeic”
- line 326 : “Pten” instead of “Pyen”
- line 348 : take off “for”
- lines 367-371 : “B” should be “A”? no section “B” described in the Figure 6 legend?
No comment in the main text about the results showed in Figure 6 A-B?
- Line 388 : “Figure 6B,C” should be “Figure 6C,D” ?
The images provided on the Figure 6C,B are quite dark and it is very hard to appreciate the co-localization. Can the authors improve them?
- Line 420 : “growth when used after versus concurrently with…. : this sentence is not clear at all. Can the authors modify it?
- line 438 : “the” should be taken off?
Author Response
Please find the responses attached.

Round 2
Reviewer 2 Report
In this revised version, the authors replied point by point in the covered letter to all the questions asked and change in many sections the main text and the figure legends according to suggestions and recommendations as required. This manuscript is really improved.
Only some minor spelling mistakes still remain regarding the "alpha" symbol following "SDF1" and on line 419 the word "alone" is written twice.